# Heteroatom-Enhanced Porous Carbon Materials Based on Polybenzoxazine for Supercapacitor Electrodes and CO_2_ Capture

**DOI:** 10.3390/polym15061564

**Published:** 2023-03-21

**Authors:** Thirukumaran Periyasamy, Shakila Parveen Asrafali, Seong-Cheol Kim

**Affiliations:** School of Chemical Engineering, Yeungnam University, Gyeongsan 38541, Republic of Korea

**Keywords:** polybenzoxazine, activation, porous carbons, electrode materials, CO_2_ adsorption

## Abstract

Through a solution method utilizing benzoxazine chemistry, heteroatoms containing porous carbons (HCPCs) were synthesized from melamine, eugenol and formaldehyde, followed by carbonization in a nitrogen atmosphere and chemical activation with KOH at three different activation temperatures, 700, 800 and 900 °C. The introduction of melamine and eugenol to the monomer produced structurally bonded nitrogen and oxygen in porous carbons. Changing the calcination temperature can alter the doping level of heteroatoms and the particle size. These carbon materials exhibit large pore size distributions, tunable pore structure, high nitrogen and oxygen contents and high surface areas, which make them suitable for use as electrode materials in supercapacitors. As a result of activating at 800 °C, the sample HCPC-800 exhibits a high specific surface area of 984 m^2^/g, high oxygen and nitrogen content (3.64–6.26 wt.% and 10.61–13.65 wt.%), hierarchical pore structure, high degree of graphitization and good electrical conductivity. An outstanding rate capability is also demonstrated, as well as incredible longevity, retaining the capacitance up to 83% even after 5000 cycles in a solution containing 1 M H_2_SO_4_. Moreover, the activated porous carbon containing nitrogen exhibits a CO_2_ adsorption capacity of 3.6 and 3.5 mmol/g at 25 °C and 0 °C, respectively, which corresponds to equilibrium pressures of 1 bar.

## 1. Introduction

During the past decade, researchers have focused their efforts on the potential applications of nitrogen-containing porous carbons for energy storage, catalysis and adsorption. Carbonaceous materials have been fine-tuned using a variety of strategies. The behavior of carbon has been extensively studied through the activation and doping processes [1,2,3]. A range of applications, including supercapacitors, fuel cells, solar cells, lithium-ion batteries, carbon capture, heavy metal adsorption, photo-catalysis and biomedicine, have demonstrated the superior performance of carefully designed carbon materials [4,5,6]. The morphology of carbon plays a critical role in all these applications. Biomedical applications are widely studied with heteroatom-doped carbon materials. There is enhancement of the surface properties of carbons that are doped with nitrogen, such as polarity and conductivity, which makes them potential candidates for energy storage [7,8,9]. Moreover, the carbon matrix containing nitrogen has been shown to selectively adsorb CO_2_ over N_2_ due to the high initial isosteric heat of CO_2_ adsorption. The nitrogen content in the carbon material determines the basicity and chemical adsorption of CO_2_ rather than determining its physical adsorption. A dramatic increase in capacitance has been demonstrated for supercapacitors when nitrogen is doped or carbon is activated. It is not yet entirely clear what mechanism(s) is responsible for the capacitance enhancement caused by nitrogen doping [10,11,12,13]. Recent studies have shown that nitrogen doping could alter the electronic structure of carbon, resulting in an increase in interfacial capacitance and charge carrier density. Carbons doped with nitrogen have been synthesized primarily by three methods: (i) for nano-casting and soft-templating processes, nitrogen-containing commercial monomers are used as starting precursors (melamine, pyrrole, acetonitrile, 1-butyl-3-methylimidazolium dicyanamide, ionic liquids and polyaniline); (ii) biomass and waste products containing nitrogen are used as starting materials; and (iii) an elevated temperature method is used to post-treat carbons with ammonia and urea. Since the first two methods are easy to synthesize and efficient in industrial processes, nitrogen-containing carbons are usually synthesized using the first two methods. Even though these methods are easy to adopt, there are several disadvantages associated with them, including less porosity, low surface area, poor thermal stability and low char yields, which result in poor CO_2_ adsorption [14,15,16,17,18].

A recent work showed that heteroatom-containing carbon with smaller pores exhibited better rate performance and higher capacitance than carbon (without heteroatoms) with high surface area. As a result of this study, it was found that both small particles and highly monodispersed particles can reduce mass transfer resistance and charge transfer resistance, resulting in improved performance of electric double-layer capacitors (EDLCs). A smaller diameter microporous carbon sphere has an advantage over a larger diameter microporous carbon material. This is because it is readily accessible to the electrolyte solution, resulting in a higher capacitance value [19,20,21].

In this regard, polybenzoxazines (PBzs) have been considered as promising precursors for the synthesis of nitrogen-rich carbons. Aside from being design flexible, they are also highly thermally stable, have a high char yield, require no catalysts or additives, produce no byproducts and produce near-zero shrinkage products following polymerization. In PBzs, a stable six-membered ring in the main chain containing nitrogen is involved in hydrogen bonding that could be maintained during thermal treatment resulting in high char yield after pyrolysis [22,23]. This paper reports on the solution method for synthesizing nitrogen-containing polymers using melamine playing two different roles, as a catalyst and as a precursor for nitrogen. These polymers become nitrogen-rich during carbonization under a nitrogen atmosphere, resulting in porous carbons containing unprecedented amounts of nitrogen. Melamine contains both pyridinic and pyrrolic nitrogens, which are chemically incorporated into the polymer and carbon structures. This kind of heteroatom-doped porous carbon is also suitable for use as CO_2_ adsorbents and supercapacitor electrode materials. The characterizations pertaining to material synthesis, CO_2_ adsorption and electrode materials for supercapacitors are well-described in this work.

## 2. Synthesis of Heteroatoms Containing Porous Carbon Sheets via Polybenzoxazine (HCPC)

In a 500 mL three-necked round-bottomed flask equipped with a reflux condenser and magnetic stirrer, formaldehyde (0.3 m, 9 g) and melamine (0.05 m, 6.3 g) were mixed together. Then 200 mL of pure ethanol was added to the mixture, and the mixture was heated to 80 °C for one hour. A solution of eugenol (24.6 g, 0.15 m) in absolute ethanol (100 mL) was added dropwise to the reaction mixture and heated at 100 °C for 12 h. Once the reaction mixture had been cooled to room temperature, 500 mL of 1 N NaOH was poured over the reaction mixture. To obtain benzoxazine powder, the precipitate was filtered and dried at 60 °C in a vacuum oven. The benzoxazine monomer is designated as EM-Bz, as it is synthesized from eugenol and melamine. Stepwise curing at different temperatures, i.e., 100, 150, 200 and 250 °C for 1 h each, was adopted to convert benzoxazine to polybenzoxazine. Following curing, the polybenzoxazine (PBz) was carbonized by heating at 600 °C for 5 h under a nitrogen atmosphere with a ramp rate of 1 °C min^−1^. Activation of the carbonized product was carried out by thorough mixing of the carbonized sample with a solution of KOH in a 1:2 ratio, followed by heating at 120 °C for 12 h. Activation was carried out in a tube furnace at three different temperatures, i.e., 700, 800 and 900 °C for 1 h with a ramp rate of 3 °C min^−1^ under nitrogen flow. It was necessary to repeatedly wash the activated products with 1 M HCl and deionized water in order to achieve a pH of neutrality after the activation process. Finally, the products were dried at 110 °C for 12 h. With respect to the activation temperatures of 700, 800 and 900 °C, the carbon materials were denoted as HCPC-700, HCPC-800 and HCPC-900, respectively. The details regarding materials used and instrumentation is given in Appendix A, respectively.

## 3. Results and Discussion

### 3.1. Structural Properties of Benzoxazine Monomer and HCPCs

By using FT-IR, ^1^H and ^13^C-NMR techniques, a characterization of the prepared benzoxazine monomer was conducted before curing. The condensation of eugenol and melamine and formaldehyde was used to synthesize EM-Bzo (Figure 1a). In the FT-IR spectrum (Figure 1b), EM-Bzo exhibits a characteristic peak at 939 cm^−1^, denoting oxazine ring structure at 1246 and 1022 cm^−1^ due to the asymmetric and symmetric vibration of C–O–C, respectively. A band can be seen at 1142 cm^−1^ for the C–N–C stretching vibrations, as well as a peak at 1372 cm^−1^ caused by tetra-substituted benzene [24]. Furthermore, methoxy carbonyl bound to the benzene ring exhibited symmetric and asymmetric vibrations at 1021 and 1247 cm^−1^, respectively. Moreover, eugenol’s alkyl chain, as well as the methylene group of its benzoxazine ring, gave bands between 2976 and 2828 cm^−1^, respectively [25]. In addition to ^1^H-NMR analysis, ^13^C-NMR analysis provided further structural insight into the synthesized EM-Bz monomer. EM-Bz is a benzoxazine-containing eugenol and melamine that can be viewed simultaneously due to its specific chemical structure. The spectra of benzoxazine synthesized in Figure 1c can be seen in this figure. As a result, benzoxazine monomers exhibit proton resonances with two singlets of equal intensity. This is because of the presence of –CH_2_ protons, located at 4.5 and 5.2 ppm with respect to N–C–Ar and N–C–O, respectively. The –OCH_3_ protons produce a singlet at 3.8 ppm. The peaks at 3.2, 5.9 and 5.0 ppm can be ascribed to protons of allyl group. Between 6.5 to 7.5 ppm the aromatic rings contribute protons. Moreover, the ^13^C-NMR spectrum shown in Figure 1d shows a characteristic carbon resonance striking the oxazine ring at 49 ppm and 79 ppm, respectively. Aromatic carbons are all found between 110 and 150 ppm, with the peak at 56 ppm associated with –OCH_3_ carbon, apart from 116, 142 and 40.4 ppm being attributed to the allyl carbon [26].

The precursor for polybenzoxazine-based heteroatoms containing carbons was synthesized through the Mannich reaction. The carbonization and activation of carbon resulted in the production of novel porous and high surface area carbons. Using Raman, FT-IR (Appendix A) and XRD analyses, it was determined that the activated carbon materials possessed structural characteristics as well as graphitic properties. The Raman spectra of HCPC-700, HCPC-800 and HCPC-900 carbon materials are illustrated in Figure 2a. It is observed that the spectra of all the carbon materials show two strong peaks, one at 1358 cm^−1^, indicating the D band, due to the A_1g_ vibration by the existence of defects; and the other at 1586 cm^−1^, indicating the G band, due to the E_2g_ vibration of the sp^2^-bonded carbon [27,28,29,30]. The intensity of the D band is high in all the samples, i.e., HCPC-700, HCPC-800 and HCPC-900, indicating the presence of numerous defects, such as nitrogen and oxygen atoms. The graphitic degree of the carbon materials depends on the ratio between the intensity of the D and G bands (I_D_/I_G_) [31,32]. The values of I_D_/I_G_ were found to be 0.98, 0.88 and 0.94, respectively, for HCPC-700, HCPC-800 and HCPC-900. As all three carbon materials have similar chemical compositions and structures, only a very small change is observed in their degree of graphitization, which could be due to the result of chemical activation. It is believed that HCPC 900, which is formed at higher thermal treatment condition of 900 °C, has a higher degree of graphitization, indicating a less disordered structure [33]. This phenomenon could be due to the loss of some nitrogen and oxygen atoms at this high activation temperature. In addition to Raman analysis, wide-angle XRD patterns are used to investigate the graphitic properties of the carbon samples. Figure 2b shows the XRD patterns of the synthesized materials, i.e., HCPC-700, HCPC-800 and HCPC-900. The XRD patterns show two different diffraction peaks at 23° and 45°, corresponding to the (002) and (100) planes of the graphitic carbon. A gradual decrease in the intensity of the peak corresponding to the (100) plane was seen for HCPC-800 and HCPC-900, indicating increased graphitization. Bragg’s equation was used to determine the d-spacing of graphitic carbon. The d-spacing was found to be 0.37 nm for all the carbon materials, suggesting that these carbon materials may result in higher capacitance than the conventional graphite when used as electrode materials in supercapacitors [34,35].

Figure 2c depicts the nitrogen adsorption isotherms on the activated carbon materials. A similar hysteresis loop can be seen in the mesopore range of all three activated carbons, HCPC-700, HCPC-800 and HCPC-900. A sharp increase in N_2_ adsorption is detected at a relative pressure under 0.1, demonstrating micropore adsorption. A firm rise in adsorption isotherms is detected with an increase in relative pressure (up to P/P_0_ = 0.9), which is caused by the adsorption of N_2_ molecules on the mesopores. Above the relative pressure P/P_0_ > 0.9, there is a sharp upswing in isotherms, indicating the existence of macropores above this relative pressure [36,37]. BET isotherms were used to calculate the specific surface of the carbon materials, which were found to be 858, 984 and 1092 m^2^/g, for HCPC-700, HCPC-800 and HCPC-900, respectively. It is observed that in severe activation conditions, such as KOH etching at 900 °C, there is a slight increase in the size of the mesopores. In addition, KOH activation at 800 and 900 °C generates macropores in HCPC-800 and HCPC-900, with sizes ranging from 50 to 100 nm (Figure 2d). It could be seen that even at this high temperature it is possible for the heteroatoms, i.e., nitrogen and oxygen, to retain their functionality, which could be observed by their high chemical and thermal stability. Generally, the thermal treatment of polybenzoxazine results in higher char yield (around 50%), which was the main reason to prepare heteroatoms containing carbon materials from a Pbz source.

As shown in Figure 3, the XPS analysis reveals the chemical state of the carbon, nitrogen and oxygen atoms within the porous carbon materials. There are three peaks on the XPS curves for the HCPC-700, HCPC-800 and HCPC-900 samples, respectively, which are probably due to the binding energies of carbon (C1s), nitrogen (N1s) and oxygen (O1s). Accordingly, the XPS results indicate that the carbon materials were prepared from benzoxazine monomer, a material containing significant levels of nitrogen and oxygen, as is evident from the XPS peaks. Based on the results of this study, it can be concluded that the nitrogen-rich porous carbon synthesized in this study is free of impurities. The XPS spectra show three prominent photoelectron peaks that are due to C, N and O energy-binding at 286.1, 398.9 and 532 eV, respectively. The deconvoluted peaks analyzing C1s, O1s and N1s peaks are depicted in Figure 3b–d. The fitting of the C1s spectrum (Figure 3b) showed four peaks as follows: the first peak at 284.6 eV is attributed to (C=C/C–C); the second peak at 285.3 eV is caused by the carbon atoms bonded with the nitrogen atom in the C-N bond; the third peak at 285.9 eV is caused by carbon atoms bonded with the O and N (HN–C=O) groups; and the fourth peak at 288.1 eV is attributed to the O–C=O/C=N/C–OH groups. As shown in Figure 3c, activated porous carbon binds different types of nitrogen, indicating the presence of a variety of nitrogen species [38,39]. An observation was made during the study which indicates that three different types of nitrogen have been observed: pyrrolic or pyridonic nitrogen at 398.4 eV; quaternary nitrogen at 400.4 eV; and oxidized nitrogen at 404.9 eV. It could be seen that in all the activated porous carbons there is a presence of pyridinic and pyrrolic nitrogen species that can be obtained only when Pbzs are used as the carbon source. Furthermore, due to their ability to electrochemically react with acidic aqueous solutions, both pyrrolic and pyridinic nitrogen species provide higher capacitances. It is important to note that nitrogen species found within carbon matrix are quaternary in nature and play an important role in electron transfer, improving the conductivity of the matrix [40,41,42]. In addition, the level of graphitization determines the conductivity of the HCPCs. As shown in Figure 3d, the O1 s spectrum is deconvoluted into two peaks corresponding to binding energies at 531.4 and 533.0 eV, indicating the presence of quinones (ph = O), phenols (C–O–C) and hydroxyl or ether groups (C–O–C), respectively, which are due to the chemisorbed oxygen atoms or water molecules. An alkaline medium can initiate reversible redox reactions in the carbon matrix due to oxygen functional groups, but these oxygen functional groups are not electrochemically active [43,44]. There are, however, quasi-reversible pseudocapacitances found when phenolic hydroxyls are reduced or deprotonated. In an acidic alkaline electrolyte, HCPCs containing phenolic hydroxyl or ether oxygen and carbonyl oxygen generate a very high level of pseudocapacitance.

As shown in Figure 4, different activation temperatures could produce different surface areas of activated carbon. The SEM images show more porous surface for all the carbon materials after activation, as depicted in Figure 4. For HCPC-700 (Figure 4a,b), the SEM images consist of many irregular micropores of varying sizes and very few mesopores, as a result of the low activation temperature (i.e., 700 °C). The structure of HCPC-800 (Figure 4c,d), which is produced at a higher activation temperature of 800 °C, is collapsed, with many voids and macropores ranging from 500 to 1000 nm in size. The SEM images of HCPC-900, produced with a much higher activation temperature (i.e., 900 °C) develops a few micropores and a higher number of macropores (Figure 4e,f). Moreover, a large number of macropores developed on the carbon surface, which resulted in thin pore walls. It is very clear from the results that optimizing the activation temperature can customize pore structure and size distribution. The FESEM images of HCPC-800 are shown along with their associated element mapping (Figure 5a–d). There are networks of carbon particles on the carbon surfaces. The particles are made up of carbon, oxygen and nitrogen atoms. Furthermore, the results of the elemental mapping indicate that nitrogen is distributed uniformly throughout the carbon matrix. Doping the carbon material with nitrogen and oxygen was confirmed by these results. By using energy-dispersive X-ray spectroscopy (EDX), it was confirmed that carbon, nitrogen and oxygen are the primary elements found in HCPC-800 (Figure 5e). TEM analysis was conducted on HCPC-800 to confirm their morphology. The TEM image showing the open-pore network of the synthesized HCPC-800 material is presented in Figure 6a–f. It provides not only continuous electron paths for electrical contact, but also shortens the diffusion pathways and accelerate ion transport [45]. Furthermore, for HCPC-800, a greater number of pores with regular arrangements was observed, which indicates that HCPC-800 contains sp^2^-bonded carbon, which is particularly conducive to electrical conductivity. From the TEM images, hierarchical porous structures are observed that are consistent with those observed with SEM. Nitrogen species are present in relatively high levels in the activated porous carbon derived from polybenzoxazine, ranging between 3.64 and 6.26 wt.%, and oxygen species are present in relatively high levels, ranging between 10.61 and 13.65 wt.%. Due to the existence of these heteroatoms, these carbon materials are more wettable (contact angle images are given in Appendix A) and electrically conductive, facilitating the access of electrolyte ions, thus improving capacitance.

### 3.2. Electrochemical Measurements

Since HCPCs are microstructurally unique, their electrochemical performance was examined in order to evaluate their performance as electrode materials. By using a three-electrode system in an aqueous electrolyte containing 1 M H_2_SO_4_ these carbons were examined. The detailed fabrication of electrode materials are given in Appendix A. The cyclic voltammetry curves of HCPC-700, HCPC-800 and HCPC-900 are shown in Figure 7a at a scanning rate of 30 mV s^−1^ for a voltage range of 0.0–0.6 V. The CV curves for all three electrodes are quasi-rectangular within 0.0–0.6 V, which indicates the materials exhibit good capacitance. The electrical double-layer contributions (EDLCs) generated by HCPC electrodes at 30 mV s^−1^ scan rates are low because of their smaller specific surface areas. It appears that pseudocapacitive reactions are occurring when the electrodes are scanned at a rate of 30 mV s^−1^. In Figure 7b, all the HCPC electrodes have linear GCD curves with a small curvature at 0.5 A g^−1^, which indicates good capacitive characteristics. The longest discharge time was observed with the HCPC-800 electrodes, followed by the HCPC-700 electrodes. Additionally, because of the existence of mesopores and macropores, electrolytes ions can easily diffuse into the pore networks at high current density, facilitating fast electrochemical reactions [46,47,48]. In addition to its high porous structure, HCPC-800 has increased pseudocapacitance as a result of its higher activation temperature at 800 °C. The broader micropore distribution and existence of macropores and mesopores in the HCPC-800 electrode, compared with HCPC-700 and HCPC-900, results in a longer discharging time. A study of HCPC-700, HCPC-800 and HCPC-900 electrodes at 0.5 A g^−1^ current density revealed specific capacitances of 229, 324 and 267 F g^−1^, respectively (Figure 7c). With its porous structure, the NRPC-800 exhibits the highest specific capacitance. A low porosity, micropores and low surface area, combined with high diffusion resistance, cause poor capacitance for HCPC-700 and HCPC-900 electrodes. Despite all the electrodes containing heteroatoms, such as nitrogen and oxygen, pseud-capacitance is induced in the electrodes due to the presence of mesopores/macropores and the availability of the carbon surface to the electrolyte ions. Due to its improved capacitance, HCPC-800 meets all the specifications. Figure 7d shows the Nyquist plots of HCPC-700, HCPC-800 and HCPC-900. All three electrodes exhibited similar Nyquist plots. Capacitive responses were observed at high frequencies as an elongated semicircle, and at low-slung frequencies as a continuing slope. Compared with HCPC-800, HCPC-700 and HCPC-900 have a shorter slope and lower diffusion resistance. Furthermore, the charge transfer resistance of HCPC-700, HCPC-800 and HCPC-900 was found to be 2.5, 1.6 and 2.1 Ω, respectively. In contrast to the two other electrodes, the NRPC-800 electrode exhibits a higher level of ionic conductivity due to the presence of micropores and macropores and increased graphitization [45,46,47,48]. The cyclic stability of HCPC-800 was evaluated in a three-electrode system using 1 M H_2_SO_4_ aqueous electrolyte. After 5000 cycles, the cyclic stability of HCPC-800 was stable with a specific capacitance of 225 F g^−1^, exhibiting excellent long-term cyclic durability of 85% (Figure 7e). HCPC-800 electrodes are durable under cycling conditions, as illustrated in Figure 7f by the integral areas around GCD curves for the 1st and 5000th cycles.

### 3.3. CO_2_ Adsorption Capacity

The bio-based benzoxazine from melamine and eugenol was used as the precursor to produce nitrogen-rich carbon materials. Generally, the CO_2_ adsorption capacity of porous materials depends on two different factors: the amount of nitrogen content and the availability of pores. This is because the nitrogen atom interacts with the CO_2_ and holds it, whereas the pores (micropores, mesopores and macropores) facilitate the diffusion and transmission of the CO_2_. As shown in Figure 8a,b the CO_2_ adsorption capacity of NRPC 700, NRPC 800 and NRPC 900 was found to be 2.6, 3.6 and 2.7 mm/g at 273 K and 2.2, 3.2 and 2.4 mm/g at 298 K, respectively. The results show that the CO_2_ adsorption capacity of NRPC 800 is high when compared with NRPC 700 and NRPC 900. Although NRPC 700, NRPC 800 and NRPC 900 have similar nitrogen contents, their different carbonization temperatures (i.e., 700, 800 and 900 °C) play an important role in determining CO_2_ adsorption capacity.

The enhanced CO_2_ adsorption capacity of NRPC 800 is due to the fact that it has hierarchical porous structure containing micropores, mesopores and macropores. This hierarchical porous structure paves the way for the fast diffusion of CO_2_ from the surface of NRPC into the active sites through chemical bonding and physical adsorption, and thus facilitates the adsorption capacity of CO_2_. However, in the case of NRPC 700 and NRPC 900, only micropores (for NRPC 700) and macropores (for NRPC 900) are present, inhibiting the diffusion of CO_2_ from the NRPC surface to the interstices, and hence decreasing the adsorption capacity of CO_2_. In both of these materials (i.e., NRPC 700 and NRPC 900), the CO_2_ adsorption capacity is due only to the nitrogen content. However, in NRPC 800, both the nitrogen content and hierarchical porous structure favors enhanced CO_2_ adsorption. The CO_2_ adsorption process takes place through three different phenomena. Firstly, the micropores present on the surface of the NRPC help in trapping the CO_2_ gas. Secondly, the nitrogen content in the NRPC, containing pyrrolic-N and pyridinic-N, helps in anchoring the adsorbed CO_2_ [49,50,51,52]. Finally, the mesopores act as a channel for the diffusion of CO_2_ from the surface to the active sites. At a higher temperature of 298 K, the CO_2_ adsorption capacity of all the NRPC materials decreased. This is probably due to the phenomenon that CO_2_ adsorption is an exothermic process and therefore it favors low temperature conditions. A similar phenomenon is observed in CO_2_ desorption at 273 and 298 K (Figure 8c,d). A comparison of CO_2_ uptake values are given in Appendix A. The above results confirm that carbon materials prepared from bio-based benzoxazines can have increased nitrogen content, which can produce stable materials with enhanced CO_2_ adsorption capacity.

## 4. Conclusions

Our study developed an easy, “one-pot” method for synthesizing heteroatom-containing polymers. Carbonizing and activating these polymers at different temperatures over a prolonged period of time results in the formation of HCPCs with improved surface properties. The HCPCs produced have a large number of fine micropores, mesopores and macropores within their structure. Furthermore, all the HCPCs have a large surface area containing a high number of nitrogen and oxygen atoms. These contribute to the excellent CO_2_ adsorption capacity and exceptional electrochemical properties of these HCPCs. The electrode fabricated from HCPC-800 exhibits a specific capacitance of 324 F/g in 1 M H_2_SO_4_ electrolyte solution. The significant contributions of these features can be attributed to a high surface area, porous structure, the presence of nitrogen and oxygen functionalities and a high degree of graphitization of the material. In addition, HCPC-800 is stable up to 5000 charge/discharge cycles, retaining superior cycling durability. The presence of small particle sizes and proper pore size distribution facilitates the penetration of the electrolyte solution into the particles. The nitrogen and oxygen functionalities increase the wettability of carbon surfaces, improving their accessibility to electrolyte solutions, thereby enhancing the accessibility of the active surface area. In addition, the nitrogen species of the pyridinic, pyrrolic and/or pyridone types makes a minor contribution to the pseudocapacitance in addition to the EDLCs, which is mainly due to the high microporosity of the carbon surface. The nitrogen-containing carbon materials also demonstrated a relatively high CO_2_ adsorption capacity of 3.6 mmol/g at 0 °C and 1 bar in terms of adsorption capacity. The obtained results confirm the promising application of bio-based Pbzs as carbon materials for supercapacitor and CO_2_ adsorption.

## Figures and Tables

**Figure 1 polymers-15-01564-f001:**
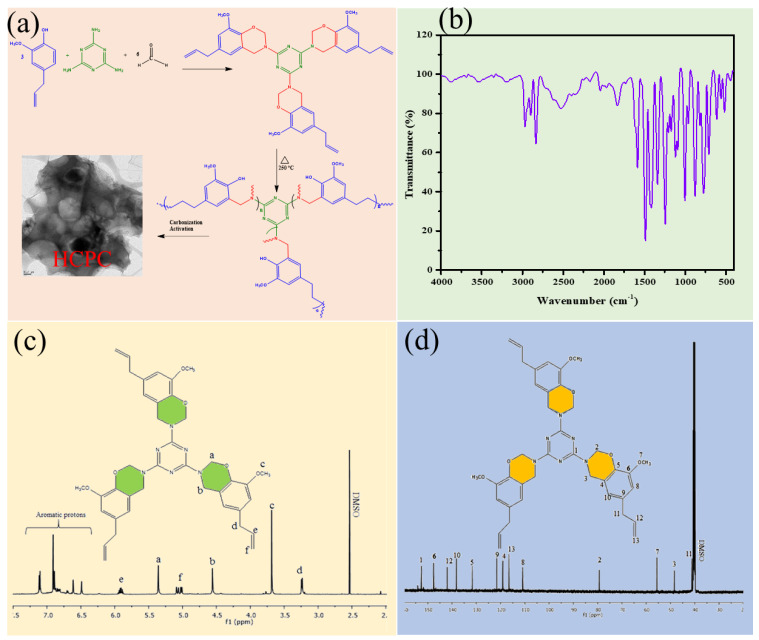
(**a**) Synthesis of (EM-Bzo & HCPC) and FT-IR, ^1^H & ^13^C-NMR spectra of benzoxazine monomer (EM-BZo) (**b**–**d**).

**Figure 2 polymers-15-01564-f002:**
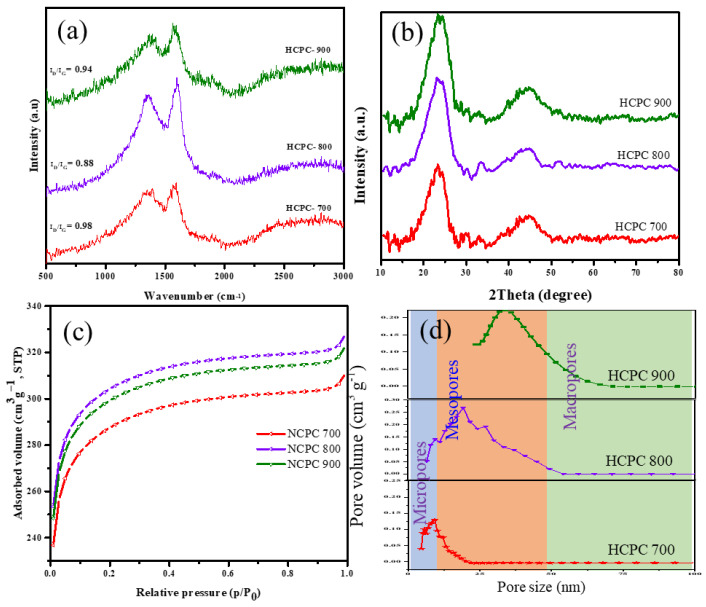
(**a**) Raman spectra, (**b**) XRD, (**c**) N_2_ adsorption isotherms and (**d**) pore size distribution of NRPCs.

**Figure 3 polymers-15-01564-f003:**
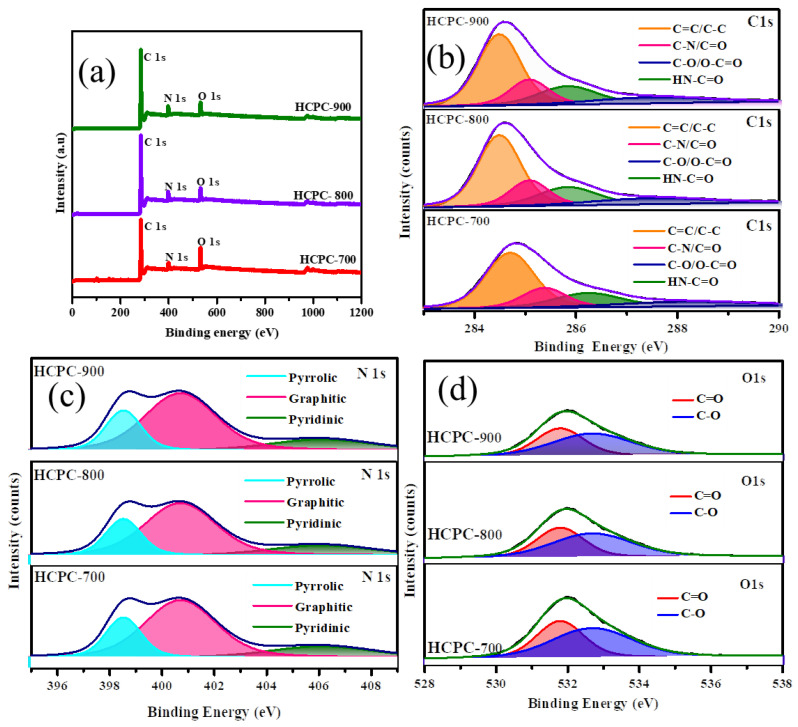
(**a**) XPS spectra and (**b**–**d**) C1s, N1s and O1s spectra of HCPCs.

**Figure 4 polymers-15-01564-f004:**
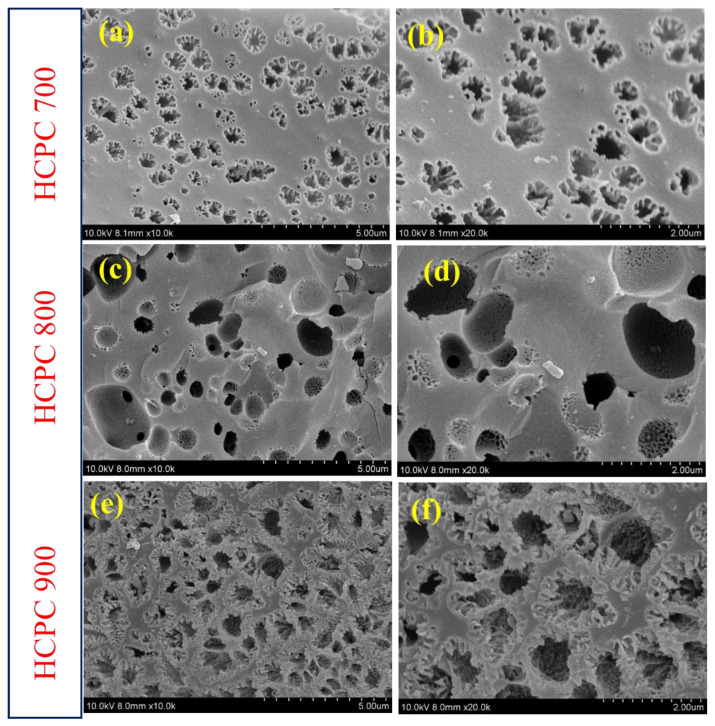
SEM images of (**a**,**b**) HCPC-700, (**c**,**d**) HCPC-800 and (**e**,**f**) HCPC-900.

**Figure 5 polymers-15-01564-f005:**
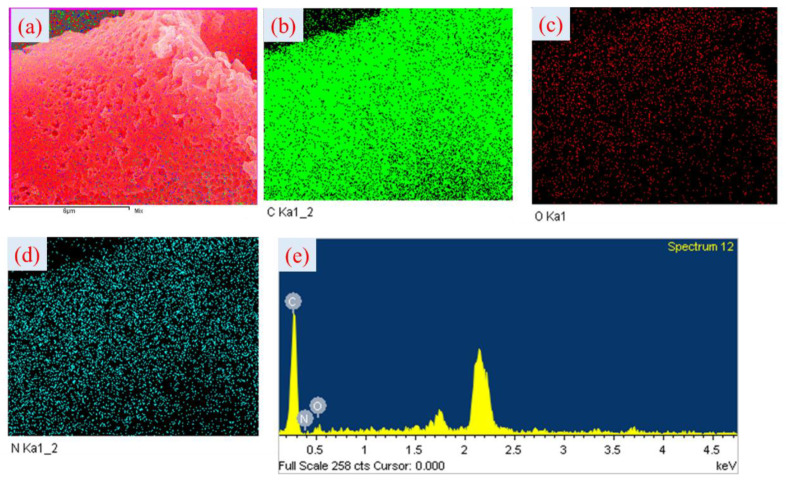
FESEM image with corresponding elemental mapping and EDX of (**a**–**e**) HPCPC-800.

**Figure 6 polymers-15-01564-f006:**
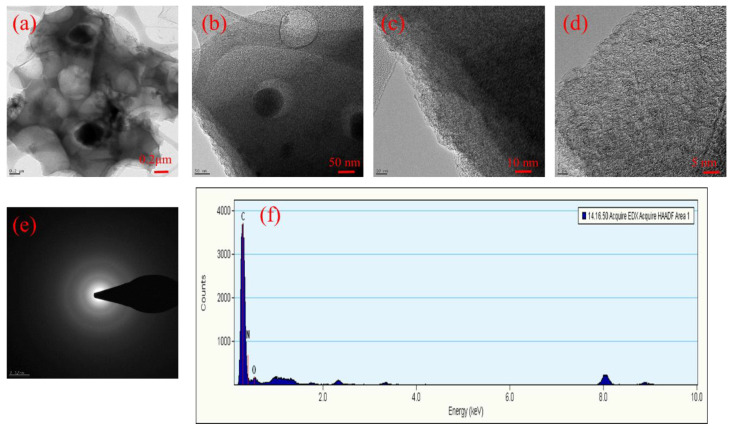
TEM image with corresponding SAED pattern and EDX of (**a**–**f**) HPCPC-800.

**Figure 7 polymers-15-01564-f007:**
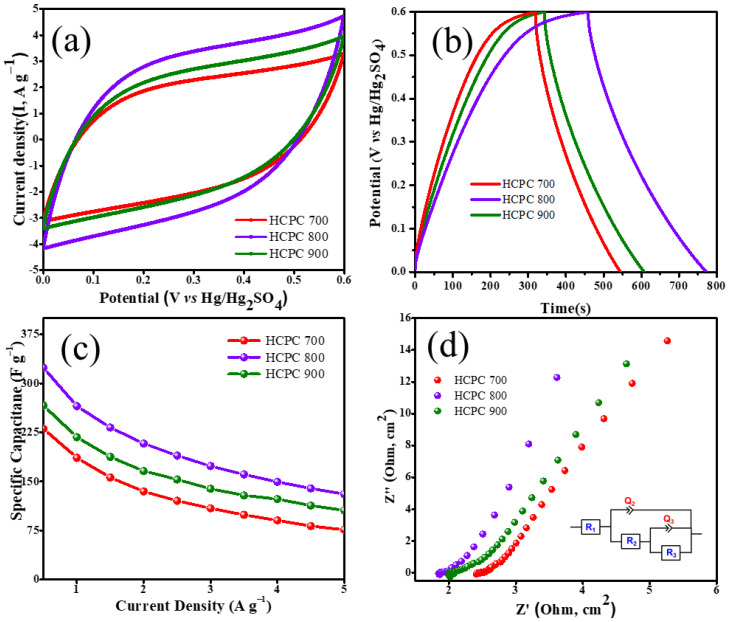
CV (**a**), GCD (**b**), Specific capacitance calculated from GCD (**c**), Nyquist plots (**d**) of HCPC-700, HCPC-800 and HCPC-900, Cycling stability of HCPC-800 at 5A/g (**e**) and GCD for 1st and 5000th of HCPC-800 (**f**).

**Figure 8 polymers-15-01564-f008:**
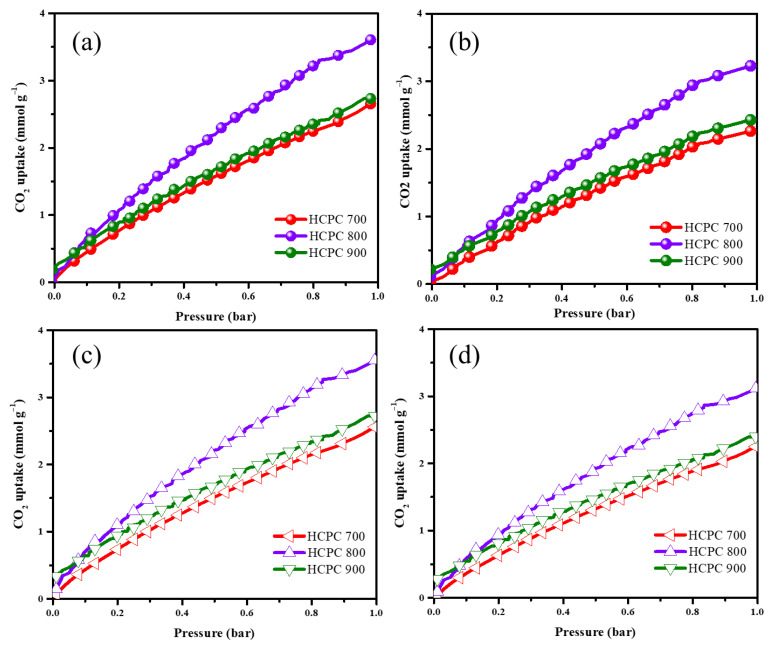
CO_2_ adsorption (**a**,**b**) and desorption (**c**,**d**) isotherms for the carbon spheres measured at 25 °C and 0 °C.

## Data Availability

The data presented in this study are available in the article.

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
