# Peer review of "Heteroatom-Enhanced Porous Carbon Materials Based on Polybenzoxazine for Supercapacitor Electrodes and CO2 Capture"

_polymers, 2023, doi:10.3390/polym15061564_

Round 1

Reviewer 1 Report

In the manuscript titled "Heteroatoms-Enhanced Porous Carbon Materials Based on Polybenzoxazine for Supercapacitor Electrodes and CO2 Capture", Periyasamy Thirukumaran and co-authors synthesized HCPC materials under different activation temperatures and investigated their electrochemical performance and CO2 capture capacity. The work is interesting in terms of the multifunctional application of described materials. I am positive about this work and would like to recommend its publication in Polymers after some revisions.

1.      The authors claimed that N and O functional groups increase the wettability of the carbon surface. Please provide some characterizations, for example, the contact angle tests.

2.      In Fig. 2d, the tick labels are very indistinct. Moreover, the definition between micropores/mesopores/macropores seems to be different from the one that is generally considered, for example, the micropores are usually below 2 nm, micropores are in between 2 and 50 nm.

3.      In the three-electrode system, the process of preparing the working electrode should be added to the text.

4.      In the CV curves, the authors mentioned that “It appears that pseudo-capacitive reactions are occurring when the electrodes are scanned at a rate of 30 mV s-1”, is this pseudo-capacitive provided by the phenolic hydroxyl or ether oxygen and carbonyl oxygen in the XPS analysis? If so, please express the possible redox reaction related to the pseudo-capacitance.

5.      Comparisons of electrochemical properties and CO2 adsorption capacity with other similar materials should be provided.

6.      The corresponding fitting circuit diagram should be inserted in Fig. 7d.

7.      The annotation of the vertical coordinate in Fig. 7a should be current density.

8.      The value of the CO2 adsorption capacity in the conclusion is different from the one reported in the main text, please check them.

Author Response

Reviewer 1

In the manuscript titled "Heteroatoms-Enhanced Porous Carbon Materials Based on Polybenzoxazine for Supercapacitor Electrodes and CO2 Capture", Periyasamy Thirukumaran and co-authors synthesized HCPC materials under different activation temperatures and investigated their electrochemical performance and CO2 capture capacity. The work is interesting in terms of the multifunctional application of described materials. I am positive about this work and would like to recommend its publication in Polymers after some revisions.

  1. The authors claimed that N and O functional groups increase the wettability of the carbon surface. Please provide some characterizations, for example, the contact angle tests.

Response: As mentioned by the reviewer, the contact angle analysis for carbon has been included in supplementary information.

  1. In Fig. 2d, the tick labels are very indistinct. Moreover, the definition between micropores/mesopores/macropores seems to be different from the one that is generally considered, for example, the micropores are usually below 2 nm, micropores are in between 2 and 50 nm.

Response: Figure 2d is replaced with correct labels showing the proper range of micro, meso and macro pores. Kindly refer Figure 2d.

  1. In the three-electrode system, the process of preparing the working electrode should be added to the text.

Response: As mentioned by the reviewer, the preparation of the working electrode has been included in the supplementary information.

  1. In the CV curves, the authors mentioned that “It appears that pseudo-capacitive reactions are occurring when the electrodes are scanned at a rate of 30 mV s-1”, is this pseudo-capacitive provided by the phenolic hydroxyl or ether oxygen and carbonyl oxygen in the XPS analysis? If so, please express the possible redox reaction related to the pseudo-capacitance.

Response: These are the possible redox reactions that occur

At anode:            2 H2O   4H+ + O2 + 4e-

At cathode:         CO2 + 2H+ + 2e-     CO + H2O

  1. Comparisons of electrochemical properties and CO2 adsorption capacity with other similar materials should be provided.

Response: A comparison table has been included in the supporting information. Kindly refer Table S1

  1. The corresponding fitting circuit diagram should be inserted in Fig. 7d.

Response: As mentioned by the reviewer, the circuit diagram has been inserted in Figure 7d. Kindly refer Figure 7d.

  1. The annotation of the vertical coordinate in Fig. 7a should be current density.

Response: The annotation of the vertical coordinate has been changed to current density. Kindly refer Figure 7a.

  1. The value of the CO2 adsorption capacity in the conclusion is different from the one reported in the main text, please check them.

Response: The correct value of CO2 adsorption capacity has been included in conclusion part.

Reviewer 2 Report

This paper reported heteroatoms-enhanced porous carbon materials based on polybenzoxazine for supercapacitor electrodes and CO2 capture. In the supercapacitor test, an outstanding rate capability is also demonstrated as well as incredible longevity, retaining the capacitance up to 83% even after 5000 cycles in a solution containing 1 M H2SO4. Moreover, the activated 21 porous carbon containing nitrogen exhibit CO2 adsorption capacity of 3.6 and 3.5 mmol/g at 25°C 22 and 0°C, respectively, which corresponds to equilibrium pressures of 1 bar. Overall, this paper is well written and organized, I suggest acceptance after minor revision.

1 A further detailed Raman analysis should be provided, especially the calculation of coherence length along ab plane.( doi.org/10.1016/j.cej.2021.133527; doi.org/10.1016/j.carbon.2022.12.072; doi.org/10.1063/1.2196057; doi.org/10.1016/j.carbon.2023.01.028)

2 The scale bar in TEM image is not clear, please replot.

3 In the Figure 7e, the Coulombic efficiency should also be offered.

4 To prove the good rate capacity, EIS tests of three carbons should be performed.

Author Response

Reviewer 2

This paper reported heteroatoms-enhanced porous carbon materials based on polybenzoxazine for supercapacitor electrodes and CO2 capture. In the supercapacitor test, an outstanding rate capability is also demonstrated as well as incredible longevity, retaining the capacitance up to 83% even after 5000 cycles in a solution containing 1 M H2SO4. Moreover, the activated 21 porous carbon containing nitrogen exhibit CO2 adsorption capacity of 3.6 and 3.5 mmol/g at 25°C 22 and 0°C, respectively, which corresponds to equilibrium pressures of 1 bar. Overall, this paper is well written and organized, I suggest acceptance after minor revision.

  1. A further detailed Raman analysis should be provided, especially the calculation of coherence length along ab plane.( doi.org/10.1016/j.cej.2021.133527; doi.org/10.1016/j.carbon.2022.12.072; doi.org/10.1063/1.2196057; doi.org/10.1016/j.carbon.2023.01.028)

Response: As suggested by the reviewer, a detailed explanation of Raman analysis have been added in the results section and the mentioned references have been included.

  1. The scale bar in TEM image is not clear, please replot.

Response: The scale bar in TEM has been replotted. Kindly refer Figure 6.

  1. In the Figure 7e, the Coulombic efficiency should also be offered.

Response: We thank the reviewer for their valuable comment. For supercapacitor applications, the discharging time from GCD curves are much important and so we calculated the capacitance retention. In our future work, we will also include the coulombic efficiency.

  1. To prove the good rate capacity, EIS tests of three carbons should be performed.

Response: We thank the reviewer for their valuable comment. For the good rate capability, we showed the GCD curves before and after cyclic stability. We will include your suggestion in our future work.

Reviewer 3 Report

In this manuscript, authors reported the preparation of heteroatoms containing porous carbons (HCPCs) from melamine, eugenol and formaldehyde followed by carbonization in nitrogen atmosphere and chemical activation with KOH for supercapacitor electrodes and CO2 3 Capture. The manuscript presents some relatively good data. I reviewed the manuscript in a critical manner and some of the comments are given below:

General comments

The manuscript might be a contribution of interest for “Polymers” and in principle within its specific scope, I believe is suitable for publication in this form. The quality of writing is good with some grammar and spelling errors here and there. The English language usage should be checked by a fluent English speaker and/or a professional language editing service.

Moreover, result interpretation is consistent, revealing the absence of experimental errors.

I recommend minor revision.

Specific comments

1.  Many studies in the state of the art used N-doped PC for the reported applications. Therefore, what is the novelty and the benefit of this work compared with those already reported? I suggest that authors highlight more the novelty in this study.

2. FTIR spectra of all PC samples should be reported.

3. What about the reusability of the produced PCs for CO2 adsorption?

Overall, a well structured study with well established work strategy.

Author Response

Reviewer 3

In this manuscript, authors reported the preparation of heteroatoms containing porous carbons (HCPCs) from melamine, eugenol and formaldehyde followed by carbonization in nitrogen atmosphere and chemical activation with KOH for supercapacitor electrodes and CO2 3 Capture. The manuscript presents some relatively good data. I reviewed the manuscript in a critical manner and some of the comments are given below:

General comments

The manuscript might be a contribution of interest for “Polymers” and in principle within its specific scope, I believe is suitable for publication in this form. The quality of writing is good with some grammar and spelling errors here and there. The English language usage should be checked by a fluent English speaker and/or a professional language editing service.

Response: The manuscript has been thoroughly checked for language including grammar and spelling errors and the sentences have been modified wherever necessary.

Moreover, result interpretation is consistent, revealing the absence of experimental errors.

I recommend minor revision.

Specific comments

  1. Many studies in the state of the art used N-doped PC for the reported applications. Therefore, what is the novelty and the benefit of this work compared with those already reported? I suggest that authors highlight more the novelty in this study.

Response: In the present work, we aimed at producing nitrogen incorporated carbons and so we chosed polybenzoxazine as a source of producing nitrogen rich carbons. By this way, the produced carbons can act both as electrode materials for supercapacitors and for CO2 adsorption. The novelty of this work is included in the last paragraph of introduction section.

  1. FTIR spectra of all PC samples should be reported.

Response: As mentioned by the reviewer, FTIR spectrum of all PC samples have been included in supplementary information. Kindly refer Figure S

  1. What about the reusability of the produced PCs for CO2 adsorption?

Response: We thank the reviewer for their valuable comment. In the present study we did not check the reusability of the produced PCs for CO2 adsorption. But, with respect to supercapacitor applications, the materials can be reused, as it shows 85% capacitance retention after 5000 cycles. We assure you that we will incorporate your suggestion in the future work.

Overall, a well structured study with well established work strategy.

All the comments given by the reviewers have been addressed accordingly. We hope for the acceptance of the manuscript at its earliest.

Round 2

Reviewer 1 Report

It is OK to me.